# Measuring Data Reconstruction Defenses in Collaborative Inference Systems

Mengda Yang[1], Ziang Li[1], Juan Wang[1] [*], Hongxin Hu[2], Ao Ren[3], Xiaoyang Xu[1] and Wenzhe Yi[1]

[1]Key Laboratory of Aerospace Information Security and Trusted Computing, Ministry of Education,
School of Cyber Science and Engineering, Wuhan University
[2]Department of Computer Science and Engineering, University at Buffalo, SUNY
[3]College of Computer Science, Chongqing University

## Abstract

The collaborative inference systems are designed to speed up the prediction processes in edge-cloud scenarios, where the local devices and the cloud system work together to run a complex deep-learning model. However, those edge-cloud collaborative inference systems are vulnerable to emerging reconstruction attacks, where malicious cloud service providers are able to recover the edge-side users' private data. To defend against such attacks, several defense countermeasures have been recently introduced. Unfortunately, little is known about the robustness of those defense countermeasures. In this paper, we take the first step towards measuring the robustness of those state-of-the-art defenses with respect to reconstruction attacks. Specifically, we show that the latent privacy features are still retained in the obfuscated representations. Motivated by such an observation, we design a technology called Sensitive Feature Distillation (SFD) to restore sensitive information from the protected feature representations. Our experiments show that SFD can break through defense mechanisms in model partitioning scenarios, demonstrating the inadequacy of existing defense mechanisms as a privacy-preserving technique against reconstruction attacks. We hope our findings inspire further work in improving the robustness of defense mechanisms against reconstruction attacks for collaborative inference systems.

## 1 Introduction

With the rapid advancement of deep neural network technology, the deployment of deep learning (DL) applications on edge devices is becoming more widespread. However, resource-constrained edge devices are often difficult to accommodate computationally intensive DL models. In addition, cloud-based DL inference applications require devices to upload users' private data to cloud service providers, which poses a serious privacy threat. Collaborative DL systems have recently emerged as a good alternative to address these challenges, where the local devices and the remote servers work together to run a complex neural network [Eshratifar et al., 2019; Banitalebi-Dehkordi et al., 2021; Li et al., 2021b]. A DL model can be split such that only its front part is executed on the local edge devices while its back part is performed on the remote cloud. Therefore, end users are not required to upload their private data to the cloud, and their privacy concerns can be addressed to some extent.

This paper focuses on *reconstruction attacks*, where existing research has shown that adversaries can easily recover sensitive data with the output of the middle layers of a DL model [He et al., 2019; Singh et al., 2021]. To address this problem, many defense approaches have been proposed for

---

[*]Corresponding author: jwang@whu.edu.cn

36th Conference on Neural Information Processing Systems (NeurIPS 2022).

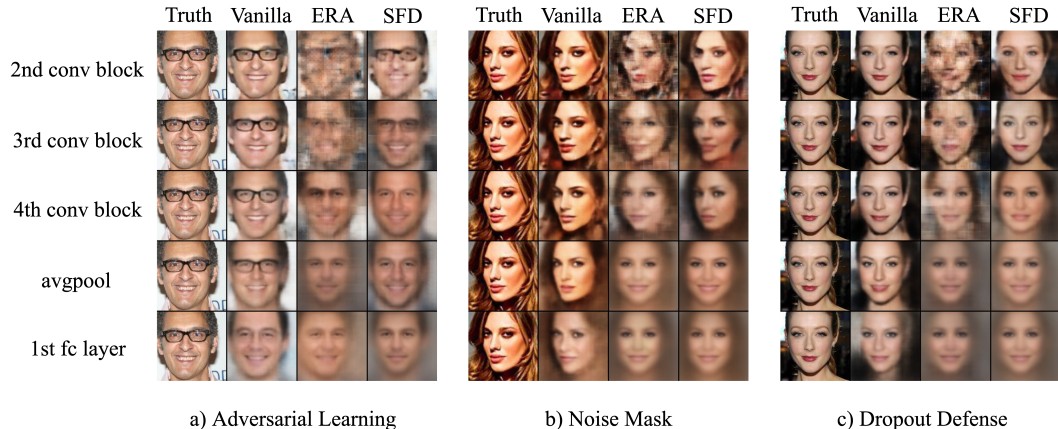

|  | Truth | Vanilla | ERA | SFD | | Truth | Vanilla | ERA | SFD | | Truth | Vanilla | ERA | SFD |

2nd conv block

3rd conv block

4th conv block

avgpool

1st fc layer

a) Adversarial Learning        b) Noise Mask        c) Dropout Defense

Figure 1: An example showing the superiority of Sensitive Feature Distillation (SFD) compared to Existing Reconstruction Attack (ERA).

collaborative inference systems. [Liu et al., 2019; Xiao et al., 2020; Li et al., 2021a] demonstrate that adversarial learning improves the robustness of the model against reconstruction attacks. [Titcombe et al., 2021; Mireshghallah et al., 2020] propose learning additive Laplace noise to protect the privacy of inference data. [He et al., 2020] uses dropout to defend against reconstruction attacks. However, none of the existing work provides a systematic security measurement for various defenses. This leads to our main research question: *Are those privacy protection techniques against reconstruction attacks for collaborative inference systems really robust enough?*

In this work, we summarise the various defenses, systematically evaluate their performance in defending against reconstruction attacks, and present their trade-offs regarding data utility and privacy leakage. We propose a novel technique to break down existing defenses, which can achieve improved reconstruction performance for various target models, datasets, and defense algorithms. We focus on the privacy of the inference data [He et al., 2019; Singh et al., 2021]. Since protectors obfuscate the intermediate feature representation to limit the redundant information about the model input, our idea is to learn to distill obfuscated features into sanitized features for better reconstruction. Specifically, we introduce Sensitive Feature Distillation (SFD) to extract the image structure knowledge of sensitive features and find that the proper formulation of prior knowledge will be helpful to solve the ill-posed reconstruction problem. We show that this vulnerability is inevitable even when different defense countermeasures are utilized. Figure 1 shows the reconstruction results using our proposed method and the existing one.

Our main contributions are summarized below:

- We are the first to experimentally verify the robustness of reconstruction defenses for inference data privacy in collaborative systems.

- We reveal the presence of latent privacy features in protected representations and validate the vulnerability of existing defense methods.

- We devise a technique called SFD against the existing defense mechanisms. SFD can achieve better reconstruction by carefully extracting sensitive information to transfer the obfuscated features into the sanitized features.

- We conduct a comprehensive measurement of the robustness of various state-of-the-art defenses leveraging our proposed technique. Our experiments show that SFD can achieve 74.2% performance improvement compared with the baseline attack, making the existing defenses significantly less effective.

# 2   Background and Related Work

This paper focuses on the most common DL task: image classification. A Deep Neural Network (DNN) model can be characterized as $F : \mathcal{X} \mapsto \mathcal{Y}$ which maps an input data $x \in \mathcal{X}$ into an output result $y \in \mathcal{Y}$.

**Collaborative Inference Systems.**   Running the entire DL inference application on the edge device can be time- and resource-consuming. A common scaling approach is to partition the DNN model where the earlier layers of the model are executed within the local edge device while the latter layers are deployed to the remote cloud [Kang et al., 2017; Eshratifar et al., 2019; Banitalebi-Dehkordi et al., 2021; Li et al., 2021b]. This scheme can reduce the computational overhead of edge devices on the one hand, and can mitigate privacy leaks of inference raw data on the other [Wang and Gong, 2018; Osia et al., 2020]. Collaborative inference empowers the deployment of DL workloads on edge-cloud platforms to be productive.

The model is split into two parts in the collaborative inference system: an on-device part and a remote cloud-based part. Each part will contain several layers of the model. From now on, we reformulate the model $F = C \circ Enc$ as the Encoder module $Enc$ and the Classifier module $C$. The edge device deploys the first part $Enc$, it feeds the users' inference data $x$ to the feature extractor to get an intermediate feature representation $z = Enc(x)$, and then sends $z$ to the cloud. The cloud manages the second part $C$, which calculates the prediction output $y = C(z)$ when the intermediate value $z$ is received from the edge device, then it returns the expected result $y$ to the edge side.

**Machine Learning Privacy and Reconstruction Attacks.**   Recent research presented a variety of privacy threats against machine learning models, such as membership inference attacks [Shokri et al., 2017a; Song and Mittal, 2021], model extraction attacks [Tramèr et al., 2016], property inference attacks [Ganju et al., 2018; Song and Shmatikov, 2019] and model inversion attacks [Yang et al., 2019; He et al., 2019]. The above threats are significant in collaborative DL systems as well.

This paper focuses on the *Reconstruction Attacks* which are closely related to much previous work on model inversion attacks. They all aim to derive private information about the original data in the context of accessing the target machine learning model. In the model inversion attack, the adversary is asked to use the association of model inputs and outputs to restore feature information from the training data. Although the original model inversion attack [Fredrikson et al., 2015] has limited performance [Hitaj et al., 2017], the inversion performance is significantly improved by training transposed CNNs on an auxiliary dataset [Yang et al., 2019; Zhang et al., 2020; Zhao et al., 2021; Wang et al., 2021].

Unlike recovering training data from the target ML models, reconstruction attacks are a type of technology designed to reconstruct model input data at the time of inference. [He et al., 2019; Singh et al., 2021] devised various attack approaches for recovering private inputs in the context of edge-cloud collaborative inference systems. In the white-box attack setting, they developed a regularized likelihood maximization algorithm, which intended to find the optimal input data for the given deep neural representations and auxiliary information. The underlying idea of the algorithm is to formulate the reconstruction attack as an optimization problem in the data space $\mathcal{X}$ to find an image $\hat{x}$ whose prediction intermediate feature representation $Enc(\hat{x})$ approximates a given $Enc(x)$. However, in order to compute the gradient to minimize the loss function, it needs white-box permission to access the target model. Furthermore, this method involves optimization for each natural sample at inference time which makes it relatively inefficient. The black-box reconstruction attack trains another neural network $G$ to recognize the mapping of the inverse from $Enc(x)$ to $x$. Specifically, without having the knowledge of the parameters of $Enc$, the adversary can collect some pairs $(z, x)$ by querying the model, and then she trains a Decoder module $G$ to learn to well approximate the mapping between intermediate representation $z$ and corresponding input $x$. In contrast to the white-box optimization-based attack, this approach is more efficient: the adversary only needs one single forward propagation through the decoder module $G$ to reconstruct any given instance. In this paper, we concentrate on the black-box reconstruction attack, which is a more practical and realistic scenario.

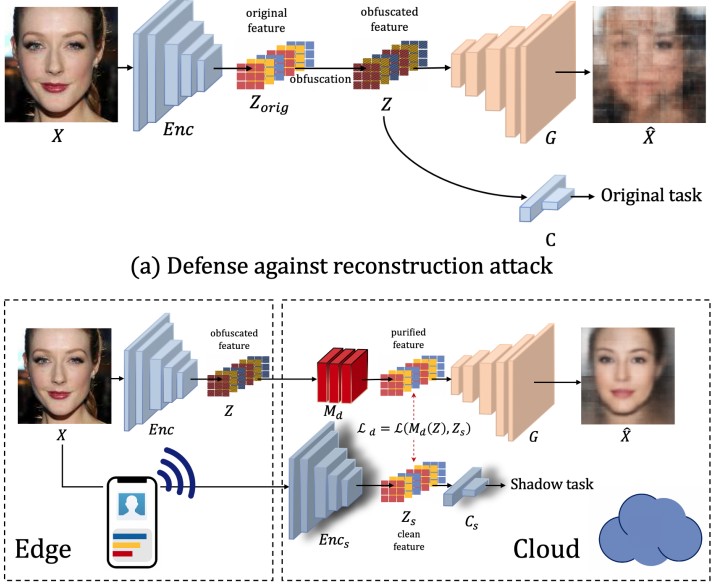

(a) Defense against reconstruction attack

(b) Reconstruction attack based on SFD

Figure 2: Reconstruction defense and attack in edge-cloud collaborative inference systems. (a) shows the main idea of defenses. (b) shows the proposed reconstruction attack based on Sensitive Feature Distillation (SFD).

# 3    Problem Setup: Defenses Against Reconstruction Attacks

## 3.1    Adversary: Learn to Reconstruct

The goal of the adversary is to reconstruct the private input data sent by the user to the edge-cloud DL inference services. Following existing works on reconstruction attacks [He et al., 2019; Singh et al., 2021], we consider a collaborative inference system between a local, edge-device part $\mathbb{E}$ and a remote, cloud-based part $\mathbb{C}$. For a partitioned model $F = C \circ Enc$, $\mathbb{E}$ performs the earlier layers $Enc$ and $\mathbb{C}$ performs $C$. We consider $\mathbb{E}$ to be trustworthy, and that $\mathbb{E}$ will properly process arbitrary input data and compute the intermediate output. $\mathbb{E}$ also attempts to prevent the input data from being reconstructed for privacy, so $\mathbb{E}$ applies the defenses to the representation and sends it to the remote part $\mathbb{C}$. The cloud $\mathbb{C}$, i.e. the adversary, is untrusted to try to steal the input.

## 3.2    Protector: Learn "NOT" to Reconstruct

For privacy, the edge part of the model typically obfuscates the intermediate feature representations before sending them to the cloud. The goal of the specially designed obfuscation is to convert the input $x$ into an intermediate representation $z$ that cannot leak sensitive features of $x$, but whose expressive power is sufficient to infer the intended label $y$. Several defense strategies have been developed to mitigate the risks of reconstruction attacks. Figure 2 (a) shows the main idea of the defenses.

**Adversarial Learning.**    The most straightforward defense against a reconstruction attack is to apply adversarial training [Goodfellow et al., 2014] to improve the robustness of the model's intermediate feature representation. [Liu et al., 2019] presents the training objective of the target model, which aims at learning a feature representation that can protect the privacy and perform normal classification.

The protector formalizes the above training objective in the following min-max optimization problem:

$$\min_{Enc,C}(\underbrace{\mathcal{L}_{task} + \lambda \ \underbrace{\max_{Dec}\mathcal{G}_{adv}}_{\text{optimal inversion}}}_{\text{optimal privacy-preserving classification}}). \tag{1}$$

, where the $Enc$ aims to fool the adversary's reconstructor model by minimizing the reconstruction gain $\mathcal{G}_{adv}$, and the performance of $C$ should be maintained. This leads to training the model with optimal privacy-preserving properties while maximizing utility.

**Noise Mask.** Noisy methods have been introduced to protect model inference [Titcombe et al., 2021; Mireshghallah et al., 2020] by adding random noise to the inference data. [Titcombe et al., 2021] suggests defending against reconstruction attacks by adding additive Laplacian noise to the feature mapping. Formally, it calculates $z^{noise} = Enc(x) + \epsilon$, where the random noise $\epsilon$ is sampled from a Laplace distribution that is parameterized by location $\mu$ and scale $b$ (In [Titcombe et al., 2021], $\mu$ is set to 0). This makes the intermediate data transferred between the edge and the cloud obfuscated by randomness every time the partitioned model is used.

**Dropout Defense.** [He et al., 2020] introduces dropout defense to defeat reconstruction attacks. Specifically, the protector generates random masks $M$ to protect the intermediate data during the inference phase of the model:

$$z^{dropout} = Enc(x) \otimes M \tag{2}$$

, where $M$ denotes a randomly generated mask, $\otimes$ denotes the element-wise multiplication. To allocate each element of $M$ to two values at random: one of them is 0 with probability $r$ and the other is 1 with probability $1 - r$. Note that the dropout defense differs from noise obfuscation in that dropout hides partially sensitive information in the feature representation, whereas noise disturbs all the information of the intermediate data.

Applying noise mask or dropout defense directly to the feature representation in prediction can introduce a significant performance loss. As a result, the protector would prefer to perform training with the noise mask or dropout defense imposed, enhancing the robustness of the model to randomness.

### 3.3 Latent Privacy Features

We refer to [Singh et al., 2021] and visualize the original face image and the learned intermediate feature representations using adversarial learning [Liu et al., 2019], as depicted in Appendix A. We can note that after obfuscation by the defense mechanism, the learned individual feature mapping retains the latent sensitive features of the input image, especially the redundant spatial feature information. Consequently, we posit that extracting the potential image structure knowledge of sensitive features can help achieve better reconstruction results.

## 4 Methodology

The main goal of this paper is to study the potential privacy leakage points of existing defenses in collaborative inference systems. Specifically, we propose an anti-defense method and describe how to exploit Sensitive Feature Distillation (SFD) for a more aggressive reconstruction attack through the carefully designed algorithm.

### 4.1 Adversary's Capability

We consider a black-box scenario where an adversary is allowed to query the target model's encoder module $Enc$ to make inferences from their collected data samples and obtain the corresponding intermediate features. For adversaries, this is a challenging attack setting [Salem et al., 2018]. The Feature Distiller module of our attack model is defined as a supervised DL model, which needs ground truth data for training. So we adopt shadow training techniques following previous works [Shokri et al., 2017b; Salem et al., 2020; Zhou et al., 2021]. We assume the adversary trains a shadow model $M_s$, which is designed to imitate the feature extraction capability of the target model. We also consider that the shadow model with the same structure as the target model. In order to achieve this

in practice, the adversary can perform model extraction attacks [Pal et al., 2020; Sha et al., 2022]. Moreover, to train the shadow model, we further assume that the adversary has an auxiliary dataset, denoted by $D_s$, that is drawn from the same distribution as the training set of the target model. Once the shadow model has been trained, the adversary can rely on it to derive the ground truth clean feature to establish her attack model. Note that most of the experiments related to shadow models' training by previous works make the same assumptions. We subsequently show in Section 6 that the assumptions regarding knowledge of the model structure can be relaxed.

## 4.2 Anti-defense Reconstruction Algorithm

Our attack scheme consists of three stages. In the first phase, the adversary constructs a shadow model locally, which is used to produce training data for the next stage. In the second phase, the Feature Distiller module of our attack model learns to distill obfuscated features into sanitized features. In the last phase, the decoder module of our attack model takes the sanitized features to produce reconstructed images. The second and third steps are performed alternately in the concrete attack flow. Figure 2 (b) provides a diagram of our attack scheme.

**Shadow Model.** The Feature Distiller module of our attack model needs to be trained in a supervised manner. Due to our minimal assumptions in Section 4.1, the adversary can only obtain the obfuscated intermediate features by querying the target model. Nevertheless, we also need ground truth data, i.e. clean intermediate features, to train the Feature Distiller. To overcome this problem, we rely on shadow training, the shadow model is applied to imitate an unshielded target model. The adversary first establishes a shadow model $M_s = C_s \circ Enc_s$ with the same structure as the target model and then uses its own dataset to train $M_s$. She can use SGD or ADAM to optimize the cross-entropy loss function of the shadow model. If the adversary's objective is to discover the inversion of the DNN for diverse tasks, then other more appropriate loss functions can be utilized. As soon as the shadow model $M_s$ is established, the adversary can utilize it to generate pure intermediate features that are not obfuscated.

**Sensitive Feature Distillation.** A straightforward reconstruction attack may not succeed if the intermediate data $z$ is obfuscated to reduce the quantity of information it can leak about the input $x$. To address this issue, we borrow the idea from the feature-based knowledge distillation [Romero et al., 2014; Zagoruyko and Komodakis, 2016]. In particular, we adapt and extend the techniques proposed in [Song and Shmatikov, 2019] to make it work in data reconstruction attacks. [Romero et al., 2014; Zagoruyko and Komodakis, 2016] use the intermediate representations of the teacher model to help the student model achieve better results in the original classification task, [Song and Shmatikov, 2019] utilizes feature representations of the auxiliary model for de-censoring to leak attributes of interest. But unlike these efforts, we use the teacher model (shadow model) to help the student model (Feature Distiller) distill sensitive information to improve the effectiveness of the reconstruction attack.

To successfully reconstruct a user's private image $x$, our approach utilizes a convolutional neural network, termed Feature Distiller, to extract missing sensitive regions in the obfuscated intermediate features. In our settings, the Feature Distiller needs to use clean feature data as the training targets. Specifically, the adversary first uses her local training set $X$ ($X$ are sampled from the adversary's datasets $\mathcal{D}_s$) to query both the target model $M$ and the shadow model $M_s$, and observes the corresponding intermediate outputs $Z = Enc(X)$ and $Z_s = Enc_s(X)$. Next, the adversary can directly train her Feature Distiller using $Z$ as the training inputs and $Z_s$ as the training targets. The feasibility of using loss on feature space for training has been demonstrated in [Romero et al., 2014; Zagoruyko and Komodakis, 2016; Song and Shmatikov, 2019], so we take Sensitive Feature Distillation as the following optimization problem to learn the Feature Distiller $M_d$:

$$\mathcal{L}_d = \sum_{z \in Z, z_s \in Z_s} \|M_d(z) - z_s\|_2^2 \tag{3}$$

. Here, we pair obfuscated features $Z$ and clean features $Z_s$ in order to encourage the Feature Distiller $M_d$ to identify the critical mappings in sensitive spatial information. After training, we can use the Feature Distiller to distill the latent vector $\hat{z}$ that is most similar to the clean feature from the target obfuscated feature $z$.

Table 1: Experiment Configurations. We generate intermediate features from the different CNNs where several convolution, batch normalization, and activation layers are assembled under a single conv block.

| Dataset | MNIST | CIFAR10 | CelebA |
|---|---|---|---|
| **Target Model** | ConvNet | VGG-16 | ResNet-50 |
| **Split point** | 1st conv layer (8*8)
2nd conv layer(4*4)
1st fc layer(64) | 1st conv block (16*16)
2nd conv block (8*8)
3rd conv block (4*4) | 2nd conv block (16*16)
3rd conv block (8*8)
4th conv block (4*4)
avgpool (1*1)
1st fc layer (512) |

**Reconstruction Attack.** At this phase, we use the same black-box attack scheme proposed in [He et al., 2019; Singh et al., 2021]: the Inverse-Network (Supervised Decoder). Notionally, the Inverse-Network is the proximate reverse function of $Enc$. However, our approach differs from the previous one in that the training input to the Inverse-Network is no longer observed data $Enc(x)$, but purified data $M_d(Enc(x))$ produced by the Feature Distiller. Meanwhile, the Inverse-Network $G$ and the Feature Distiller $M_d$ are jointly trained, which leads to better reconstruction quality.

Similar to the previous phase, the adversary gathers a stack of training samples of labeled $(z, x)$ pairs. She can then use $M_d(z)$ as the sanitized latent vectors of $z$ to train the Inverse-Network. We leverage the L2 norm in the pixel space as the loss function of $G$:

$$\mathcal{L}_G = \sum_{z \in Z, x \in X} \|G(M_d(z)) - x\|_2^2 \tag{4}$$

, $G$ is optimized to reconstruct data from sanitized latent vectors. Notably, the structure of $G$ is not necessarily correlated with the target Encoder module $Enc$. After $G$ and $M_d$ are trained, the adversary can feed any observed intermediate-level value $z^\star = Enc(x^\star)$ to the attack model and obtain the reconstructed sensitive input $\hat{x}^\star = G(M_d(z^\star))$.

## 5 Experiment

We systematically measure the robustness of existing methods for defending against reconstruction attacks and the effectiveness of our proposed scheme. We carry out extensive experiments on three representative image classification tasks, each of which uses a different architecture of the target DNN model.

### 5.1 Experiment Setup

**Datasets.** To perform experimental evaluation, we use three standard benchmark image recognition datasets: MNIST [LeCun, 1998], CIFAR10 [Krizhevsky et al., 2009] and CelebA [Liu et al., 2015]. These datasets also serve as baseline datasets for a variety of ML security and privacy tasks. In most of our experiments, we split the dataset into three parts, target models' private dataset $\mathcal{D}_p$, adversary's shadow dataset $\mathcal{D}_s$ and the test set $\mathcal{D}_t$ with the rest. We detail each dataset in the following: MNIST contains 70,000 grey-scale images. We set $\mathcal{D}_p$, $\mathcal{D}_s$ and $\mathcal{D}_t$ to 50,000, 10,000 and 10,000 respectively; CIFAR10 consists of 60,000 color images. We separate $\mathcal{D}_p$, $\mathcal{D}_s$ and $\mathcal{D}_t$ to 40,000, 10,000, and 10,000 respectively; CelebA is a dataset composed of 202,599 images of celebrities. For simplicity, the task is to predict gender. The $\mathcal{D}_p$, $\mathcal{D}_s$ and $\mathcal{D}_t$ are set to 101,299, 81,040 and 20,260 accordingly.

**Models.** For MNIST, We experiment with ConvNet which is similar to the LeNet [LeCun et al., 1998] architecture, only making the network layer wider. For CIFAR10, we apply VGG-16 [Simonyan and Zisserman, 2014]. For CelebA, we adopt ResNet-50 [He et al., 2016]. We split each target model into different layers. For Inverse-Network $G$, following [He et al., 2019; Yang et al., 2019], we use transposed CNN blocks[2] to decode target intermediate feature representations to pixel images, where each block has a transposed convolutional layer followed by a batch normalization layer as well as an

---

[2]The official implementation of [He et al., 2019]: `https://github.com/zechenghe/Inverse_Collaborative_Inference`

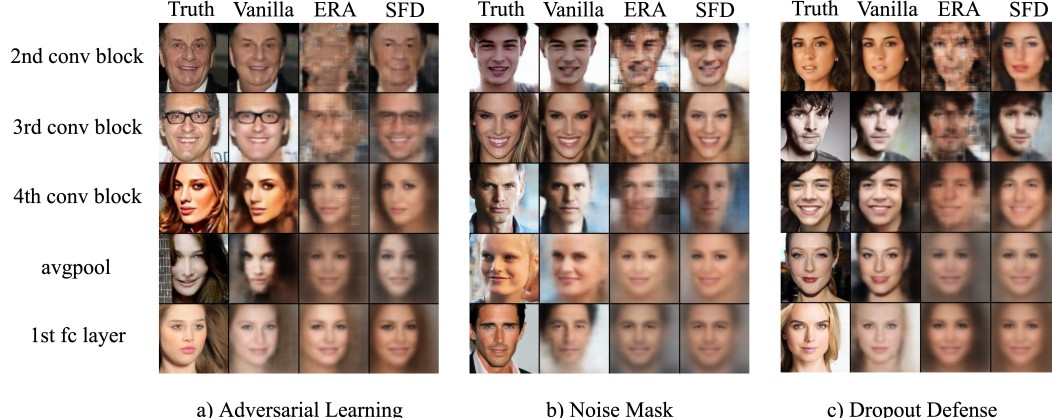

|  | Truth Vanilla ERA SFD | Truth Vanilla ERA SFD | Truth Vanilla ERA SFD |
| --- | --- | --- | --- |
| 2nd conv block | | | |
| 3rd conv block | | | |
| 4th conv block | | | |
| avgpool | | | |
| 1st fc layer | | | |
| | a) Adversarial Learning | b) Noise Mask | c) Dropout Defense |

Figure 3: Reconstruction attack qualitative evaluation: We show the reconstructed images on CelebA. Each row corresponds to different split points of ResNet-50. In both (a), (b) and (c), the first column depicts the original images, and the second column depicts the result derived from attacking unprotected features, which is the upper bound for our approach. The third column presents the reconstruction results of ERA, while the fourth column presents the results of SFD.

activation layer. The number of transposed CNN blocks increases with the deepening of the split point. For Feature Distiller $M_d$, we choose a neural network using multiple convolutional layers (when the split layer is conv block) or full connection layers (when the split layer is fc layer), with the number of hidden units set to the dimension of the feature data.

**Compared Baselines.** In the collaborative inference system scenario, we evaluate the effectiveness of the existing defenses by comparing the proposed attack with Existing Reconstruction Attack (ERA) [He et al., 2019] against all three defense methods.

**Experiment Details and Evaluation Metrics.** Table 1 gives details of the experimental configurations. To quantify the privacy risks of reconstruction attacks, we follow the setting in most of the previous work [He et al., 2019; Singh et al., 2021]. We adopt two metrics, Mean Squared Error (MSE) and Structural Similarity Index (SSIM) [Wang et al., 2004]. We use different evaluation metrics between the reconstructed data and the original instance to gather multiple pieces of evidence of how effective the attack recreates the private target sample.

### 5.2 Empirical Results and Performance Analysis

**Reconstruction Performance.** Table 2 shows the performance of the proposed approach. We report the difference values (values in brackets) between the MSE loss of reconstruction attack based on SFD and the baseline ERA, and we also report the difference values between the SSIM of the reconstructed image generated from our method and from the baseline, both of which represent performance gains. Averagely, the proposed method provides a great improvement in reconstruction results against all three defenses. We owe this to SFD used in our method, which can capture sensitive spatial information for more aggressive attacks. Note that the vanilla results show the reconstruction effect of using ERA when the target model is without defense. Here, we just use vanilla results to show the best possible results for the reconstruction. The observation is that our approach still provides a competitive performance. Specifically, on the CelebA dataset, we improve on decreasing the MSE loss from **0.124** to **0.032** and increasing the SSIM from **0.370** to **0.618** compared to ERA when the split point is set to the 2nd conv block and against adversarial learning, which brings a **74.2%** performance improvement (in terms of MSE loss values) while being close to the vanilla SSIM results **0.785**. For noise mask and dropout defense, we can achieve a **47.7%** and **40.9%** performance improvement, respectively.

Table 2: Reconstruction results against all three defenses, the data in brackets represents the attack performance improvement of reconstruction attacks based on SFD compared to the baseline ERA. Vanilla results show the reconstruction effect of using ERA when the target model is without defense. Defense hyper-parameters are set $\lambda = 5.0$ for adversarial learning, $b = 10.0$ for noise mask, and $r = 0.9$ for dropout defense. Both noise mask and dropout defense perform training with the defense algorithm imposed. The smaller the MSE and the larger the SSIM indicates the better the reconstruction results.

| Dataset | Split Point | Vanilla | | Adversarial | | Noise | | Dropout | |
|---|---|---|---|---|---|---|---|---|---|
| | | MSE | SSIM | MSE↓ | SSIM↑ | MSE↓ | SSIM↑ | MSE↓ | SSIM↑ |
| **MNIST** | 1st conv layer | 0.002 | 0.994 | 0.442-(0.310) | 0.123+(0.451) | 0.323-(0.070) | 0.138+(0.082) | 0.022-(0.006) | 0.943+(0.013) |
| | 2nd conv layer | 0.057 | 0.838 | 0.324-(0.191) | 0.109+(0.494) | 0.255-(0.089) | 0.146+(0.319) | 0.231-(0.110) | 0.299+(0.358) |
| | 1st fc layer | 0.103 | 0.709 | 0.199+(0.016) | 0.369-(0.066) | 0.140+(0.007) | 0.585-(0.036) | 0.235-(0.002) | 0.246-(0.006) |
| **CIFAR10** | 1st conv block | 0.006 | 0.942 | 0.238-(0.094) | 0.309+(0.181) | 0.045-(0.022) | 0.723+(0.116) | 0.075-(0.030) | 0.537+(0.116) |
| | 2nd conv block | 0.026 | 0.762 | 0.233-(0.045) | 0.149+(0.123) | 0.112-(0.024) | 0.409+(0.077) | 0.173-(0.044) | 0.217+(0.106) |
| | 3rd conv block | 0.100 | 0.397 | 0.230-(0.021) | 0.143+(0.023) | 0.212-(0.005) | 0.130+(0.019) | 0.223-(0.016) | 0.132+(0.035) |
| **CelebA** | 2nd conv block | 0.010 | 0.785 | 0.124-(0.092) | 0.370+(0.248) | 0.172-(0.082) | 0.289+(0.211) | 0.088-(0.036) | 0.416+(0.175) |
| | 3rd conv block | 0.020 | 0.698 | 0.189-(0.045) | 0.331+(0.121) | 0.206-(0.038) | 0.332+(0.083) | 0.155-(0.021) | 0.370+(0.090) |
| | 4th conv block | 0.038 | 0.625 | 0.216-(0.023) | 0.369+(0.048) | 0.219-(0.010) | 0.364+(0.034) | 0.202-(0.005) | 0.375+(0.027) |
| | avgpool | 0.067 | 0.551 | 0.187-(0.003) | 0.405+(0.015) | 0.231-(0.002) | 0.372+(0.015) | 0.230-(0.008) | 0.371+(0.025) |
| | 1st fc layer | 0.124 | 0.452 | 0.216+(0.005) | 0.399-(0.002) | 0.232+(0.002) | 0.376+(0.010) | 0.231-(0.003) | 0.382+(0.007) |

Table 3: Evaluation for the impact of different defense hyper-parameters on CIFAR10 dataset.

| Defense | Param | 1st conv block | | 2nd conv block | | 3rd conv block | |
|---|---|---|---|---|---|---|---|
| | | Acc | SSIM | Acc | SSIM | Acc | SSIM |
| **Adversarial** | 1.0 | 83.1 | 0.945+(0.005) | 83.5 | 0.380+(0.117) | 83.4 | 0.153+(0.014) |
| | 2.0 | 79.8 | 0.457+(0.221) | 81.8 | 0.167+(0.164) | 81.5 | 0.156+(0.017) |
| | 5.0 | 76.5 | 0.309+(0.181) | 79.0 | 0.149+(0.123) | 77.3 | 0.143+(0.023) |
| **Noise** | 1.0 | 85.5 | 0.876+(0.053) | 86.9 | 0.729+(0.019) | 86.0 | 0.283+(0.029) |
| | 5.0 | 83.7 | 0.796+(0.089) | 85.5 | 0.526+(0.062) | 85.9 | 0.144+(0.015) |
| | 10.0 | 82.3 | 0.723+(0.116) | 84.3 | 0.409+(0.077) | 84.9 | 0.130+(0.019) |
| **Dropout** | 0.5 | 84.9 | 0.818+(0.058) | 85.4 | 0.594+(0.022) | 84.9 | 0.256-(0.015) |
| | 0.7 | 83.7 | 0.732+(0.089) | 84.2 | 0.461+(0.061) | 85.8 | 0.169+(0.027) |
| | 0.9 | 79.9 | 0.537+(0.116) | 81.9 | 0.217+(0.106) | 84.6 | 0.132+(0.035) |

**Comparison for Different Split Points.** For different split points of the target model, we conduct both quantitative (Table 2) and qualitative analysis (Figure 3). The visual results show that SFD achieves remarkably better reconstruction performance than ERA, and it works better at early layers than at later layers. This is confirmed by the quantitative results where SFD obtained MSE improvement of **0.092** (**74.2%**) and SSIM improvement of **0.248** in the 2nd conv block, compared to **0.003** (**1.6%**) and **0.015** in the avgpool. It is consistent with the intuition that features are becoming more and more abstract during the forward propagation process of the neural network, so the quality of reconstructed images of both SFD and ERA deteriorates as the layer goes deeper. See Appendix C for more examples and quantitative results.

Besides, we also observe a significant deterioration in reconstructed image quality on deeper layers (especially the fully connected layer). This is due to the size of feature mapping on deeper layers is typically smaller than on shallower layers, and the fully connected layer even mixes up previous visual information, keeping only the semantic information. So it is relatively harder for SFD to distill sensitive features from small obfuscated representations. When choosing the partition point in a Privacy-Aware collaborative inference system, we suggest that the split layer on the edge device should be set deep enough (e.g., at least one fully connected layer [He et al., 2019, 2020]) to reduce the impact of reconstruction attacks.

**Effect of Defense Hyper-parameters.** Table 3 presents the reconstruction performance for varying defense hyper-parameters. The results indicate that even if the hyper-parameters of adversarial learning, noise mask, and dropout defense are set to 5.0, 10.0, and 0.9 (These parameter settings have incurred considerable utility loss), SFD can achieve SSIM improvements of 0.181, 0.116 and 0.116 respectively. This demonstrates that SFD can still effectively extract sensitive information from feature spaces that are protected by stronger defenses (more heavily obfuscated features).

In addition, we note that the attack performance of ERA decreases (lower SSIM) with increasing hyper-parameters, which will negatively affect the reconstruction results of SFD (more generic reconstructed images). Thus, even against SFD attacks, larger defense hyper-parameters could

Table 4: Performance of shadow models for a different structure. The accuracy of shadow models using VGG-16, VGG-13, and ResNet-18 is 72%, 65%, and 61%, respectively.

| Shadow Model | 1st conv block | | 2nd conv block | | 3rd conv block | |
|---|---|---|---|---|---|---|
| | MSE | SSIM | MSE | SSIM | MSE | SSIM |
| None | 0.220 | 0.457 | 0.221 | 0.167 | 0.219 | 0.156 |
| VGG-16 | 0.084 | 0.678 | 0.152 | 0.331 | 0.203 | 0.173 |
| VGG-13 | 0.082 | 0.682 | 0.154 | 0.330 | 0.206 | 0.202 |
| ResNet-18 | 0.078 | 0.671 | 0.153 | 0.302 | 0.194 | 0.207 |

potentially provide stronger obfuscation effects in reconstruction, but as a trade-off, this will severely degrade the utility of the target model. For more details on the discussion of the comparison of defense methods, see Appendix B.

## 6 Discussion

**Relaxing Attacker Model Assumption.** We explore further relaxing the attacker's knowledge of the target model structure. For simplicity, we employ a shadow model with intermediate feature representations of the same size as the target model, but with a different structure. Table 4 depicts the evaluation results. In our experiments on CIFAR10, we apply the VGG-16 described previously in Section 5.1 to the target model, and use VGG-16, VGG-13, and ResNet-18 for the shadow models respectively. The results of the reconstruction of the intermediate features of different sizes show both worse and better performance, but the variations are small, indicating that SFD is robust against such changes in the architecture of the shadow model.

**Possible Defenses.** As shown in Table 2 and Figure 3, both quantitative and qualitative measurements of the reconstruction images become more generic as the split point becomes deeper. Meanwhile, increasing the defense hyperparameters can help mitigate the attacks to a degree, as illustrated in Table 3. Combining these two strategies may provide better privacy protection for inference data. However, this may cause a heavy computational burden on the client devices and reduce model accuracy. We have made it our future work to explore mechanisms to effectively defend against reconstruction attacks in collaborative inference systems.

## 7 Conclusion

In this work, we track the potential privacy leakage points of defense mechanisms against reconstruction attacks in collaborative inference systems. We posit that extracting the latent image structure knowledge of sensitive features can help achieve better reconstruction. Therefore, we devise an anti-defense method using the sensitive feature distillation (SFD) technique to bypass the existing defense mechanisms. SFD is designed to purify feature representations that are obfuscated by the defense mechanisms, so as to better extract sensitive information. We systematically measure the effectiveness of our method against three state-of-the-art defenses. Experimental results show that SFD achieves remarkable performance improvements over existing reconstruction techniques on several datasets, which demonstrates that current defense schemes are not robust enough. We hope our findings encourage critical exploration of defense mechanisms against reconstruction attacks for private collaborative inference.

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
