# 8   Appendix

## A   Visualization of Latent Privacy Features

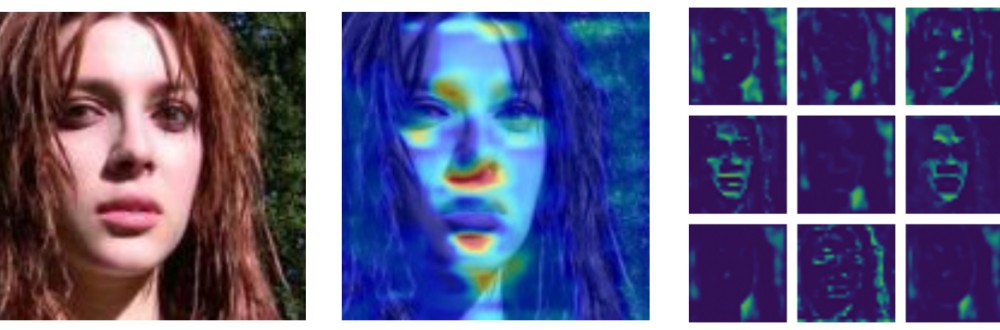

Original Image          Grad-CAM Visualization     Random Features Visualization

Figure 4: Feature visualization using Grad-CAM and the corresponding randomly selected individual feature mapping. These learned feature mappings are derived from the ResNet-50 applying adversarial learning defense on CelebA.

## B   Comparison of defense methods

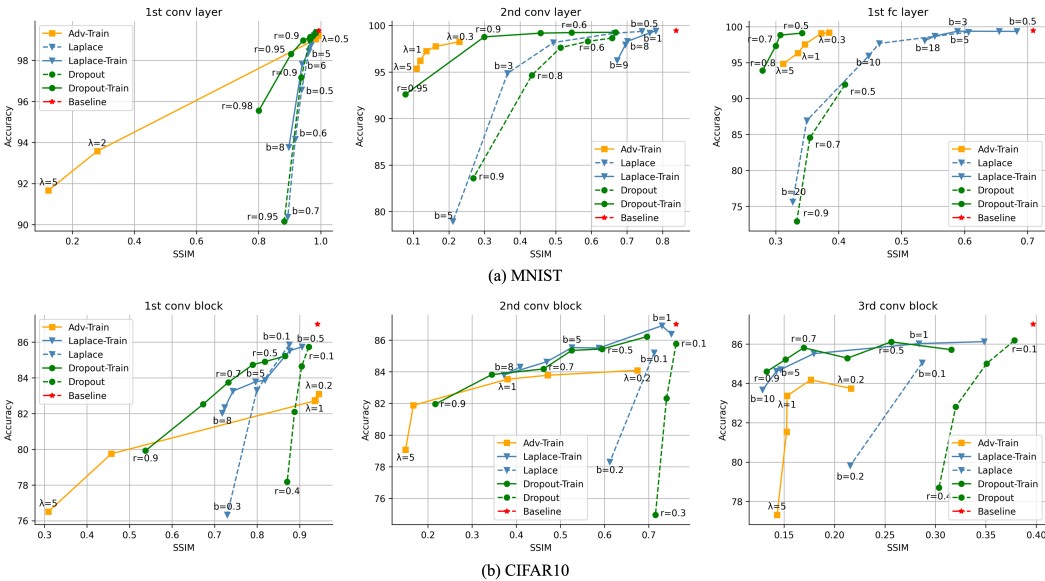

Figure 5: Utility-Privacy trade-off of different defenses against existing reconstruction attack. (a) shows the result of the MNIST dataset. (b) shows the result of the CIFAR10 dataset.

Through performing the ERA, we verify the performance of defenses. Figure 5 shows the quantitative analysis of data utility and recovered image quality on MNIST and CIFAR10 datasets and partitioned layers of different target models, respectively. Greater accuracy means improved usability, whereas lower SSIM represents better privacy. We observe that larger defense hyper-parameters lead to a more significant loss of model accuracy and thus better privacy-preserving, demonstrating the trade-offs between utility and privacy.

**Adversarial Learning.** The adversarial learning is represented as orange lines. The $Enc$ trained with this method maintains a good task performance with a decrease in the inverse image quality. However, this method also results in a more significant loss of model accuracy for better privacy-preserving. Our results suggest that defending reconstruction attacks with adversarial learning may require the regularization factor $\lambda \geq 5.0$. As a trade-off, such a large factor would introduce the accuracy loss of around $5\%$ for MNIST and $10\%$ for CIFAR10.

**Noise Mask.** The noise mask with training and adding noise directly to the intermediate feature is represented as solid and dotted blue lines. The model accuracy degradation can be mitigated to some extent by using noisy training instead of adding noise directly, however, the reconstruction is still recognizable (SSIM > 0.3 [He et al., 2019, 2020]). This suggests that noise may not be a practical defense to achieve the desired protective effect.

**Dropout Defence.** We use the green solid line to indicate the dropout defense with training, and follow with the green dotted line to indicate the dropout defense applied directly. Similar to noise obfuscation, adding dropout masks to the target layer during the training phase provides better privacy. We also find that dropout defense is superior to the other two defense options at the deeper layers. Dropout exploits the parameter redundancy of the DNN model (a fully connected layer would introduce greater redundancy) to remove sensitive information, while the other two approaches cause greater accuracy loss by disrupting clean intermediate features.

## C  More CIFAR10 Examples

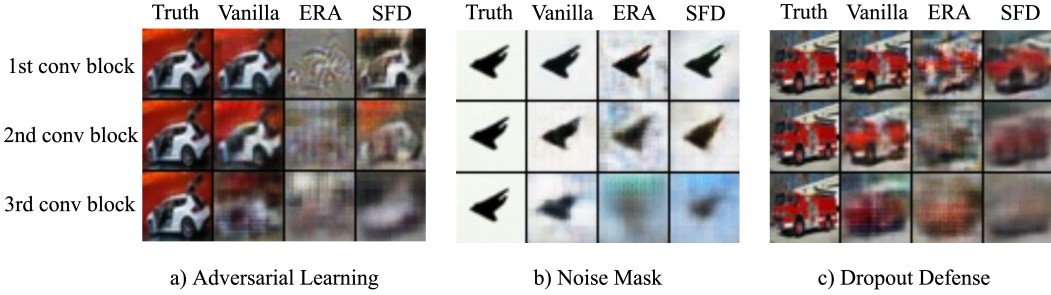

Figure 6: Additional qualitative CIFAR10 examples.