# OpenReview forum: "Measuring Data Reconstruction Defenses in Collaborative Inference Systems"
_NeurIPS.cc/2022/Conference — NeurIPS 2022 Accept_

### Official Review · Reviewer_ihvh · 2022-07-11

**Rating:** 7
**Confidence:** 4
**Soundness:** 4 excellent
**Presentation:** 3 good
**Contribution:** 4 excellent

**Summary:**

This paper proposes an attack model for edge-cloud collaborative inference systems. More specifically, the authors develop a technique for evaluating defense mechanisms against model inversion attacks, where the attacker uses obfuscated features (computed on a trusted local edge device and to be fed to an untrusted cloud service for prediction) to recover users’ private data (i.e., the raw image). To this end, the authors propose a technique called Sensitive Feature Distillation (SFD) to sanitize obfuscated features generated by a defense technique, which are then fed to an Existing Model Inversion (EMI) attack to restore the original image. The proposed technique is evaluated against three defense techniques (adversarial learning, noise mask, dropout) on three datasets (MNIST, CIFAR10, CelebA), where it is shown to outperform a baseline EMI attack (owing to the SFD model).

**Questions:**

- I'm interested to see if an SFD trained on one dataset can transfer to another. This would further relax the constraints on the attacker since they don't always have access to data with the same distribution as the training dataset. Have the authors performed any such experiments, or do they have any insight on the robustness of the SFD in this setting?
- Figure 3b, the reconstructed images look more and more like generic images as you go down the rows. Is this also observed on other examples? Perhaps the authors could show some examples with the same image in all rows in the appendix.

**Limitations:**

The proposed method has very few constraints as it only requires blackbox access to the target model, and access to data with the same distribution as the training dataset for training the shadow model. This is discussed in good detail in the paper.

**Strengths And Weaknesses:**

I enjoyed reading this paper. The authors have developed a novel technique that can be used for evaluating defense techniques in edge–cloud collaborative inference systems. The proposed method is well-described and technically sound, and the black box attack setting makes this applicable to a wide range of scenarios. The evaluation results also reveal that current defense mechanisms cannot adequately defend against adversarial attacks, similar to many other domains.

One weakness of this type of inversion attack is that the recovered image tends to look more like a generic image of the same class (see the Figure 3b, specially the bottom rows) rather than the original image, and the used metrics somewhat fail to capture this distinction. So perhaps the privacy issue is less severe than what the actual numbers suggest. Nevertheless, the proposed technique is novel and succeeds in outperforming the baseline, which makes it a solid contribution for evaluating defense techniques.

Minor comments:
- Line 315: "Strong defence does not help, the performance of our attack increases along with the hyper-parameter increases", isn't the performance of the attack decreasing (lower SSIM) with larger hyperparameters?
- Line 301: defense -> defence, for consistency! :)
- Line 328: "When repeating the experiments on other datasets or other", typo?
- Section numbers in the checklist are unresolved.

---

> ### Author Response · Authors · 2022-08-02
> **Response to Reviewer ihvh**
>
> We greatly appreciate the invaluable and constructive comments from the reviewer. We address each of the concerns below.
>
> **Question1:** *“I'm interested to see if an SFD trained on one dataset can transfer to another. This would further relax the constraints on the attacker since they don't always have access to data with the same distribution as the training dataset. Have the authors performed any such experiments, or do they have any insight on the robustness of the SFD in this setting?”*
>
> **A:** We have already performed such experiments. If SFD is trained with differently distributed datasets, e.g. using CIFAR10 to train the SFD, and then using that SFD to attack CelebA, the reconstruction results are not good. This is consistent with the observation of [1] and [2]: the closer the attacker’s data distribution is to the target model's training data distribution, the better the inversion quality is (both visually and quantitatively). We will add details of such experiments in our manuscript.
>
> **Question2:** *“Figure 3b, the reconstructed images look more and more like generic images as you go down the rows. Is this also observed on other examples? Perhaps the authors could show some examples with the same image in all rows in the appendix.”*
>
> **A:** Yes, this is also observed on other examples. As discussed in L303-311 of our paper, because features become more and more abstract as the network layer goes deeper, the reconstructed image quality of both EMI and SFD are deteriorated. We thank the reviewer for this constructive suggestion and will show some examples in the revised version of our manuscript.
>
> **Weakness:** *“One weakness of this type of inversion attack is that the recovered image tends to look more like a generic image of the same class (see the Figure 3b, specially the bottom rows) rather than the original image, and the used metrics somewhat fail to capture this distinction. ”*
>
> **A:** Please refer to **Question2**.
>
> **Minor comments:**
>
> **A:** First, we would like to thank the reviewer again for making these minor comments. Here are our responses:
>
> 1. We agree it. We will rephrase the sentence to: "Strong defence does not help, our attack can still achieve a good improvement".
> 2. Thanks for catching this inconsistency. We will fix it.
> 3. We will fix this typo in the revised version of our manuscript and we will delete "or other".
> 4. We will resolve this issue in the revised version of our manuscript.
>
> *References*:
>
> [1] He, Z., Zhang, T., & Lee, R. B. (2019, December). Model inversion attacks against collaborative inference. In Proceedings of the 35th Annual Computer Security Applications Conference (pp. 148-162).
>
> [2] Yang, Z., Zhang, J., Chang, E. C., & Liang, Z. (2019, November). Neural network inversion in adversarial setting via background knowledge alignment. In Proceedings of the 2019 ACM SIGSAC Conference on Computer and Communications Security (pp. 225-240).

---

### Official Review · Reviewer_d4JB · 2022-07-13

**Rating:** 4
**Confidence:** 5
**Soundness:** 2 fair
**Presentation:** 3 good
**Contribution:** 1 poor

**Summary:**

Existing works in ARL have developed different types of techniques for protecting sensitive information like sensitive attribute and sensitive input. For the sensitive input category, this authors develop an attack that can undermine the security of the existing defense techniques. Motivated by this observation, they introduces a new defense technique that perform favorably on their attack algorithms.

**Questions:**

How is your contribution 1  (L53-55) different from contribution 3 (L59-63)? I believe both refer to the experimental evaluation only.

**Limitations:**

The authors may wanna highlight limitations of their empirical evaluation but I do not see any major concern

**Strengths And Weaknesses:**

Strength -
- Sensitive feature distillation is an interesting and new idea in the ARL community.
- The experimental data highlights the vulnerability of the ARL approaches to protect inversion of inference data.

Weakness -
- The paper shouldn't call the attack techniques as model inversion since the goal is to infer sensitive information from the data used for prediction query and not the training data used.
- Wrong characterization of early MI attacks by Fredrikson et al. 2014 and Fredrikson et al. 2015 that extract sensitive attributes (Fredrikson et al. 2014) and reconstruction of original data (Fredrikson et al. 2015).
- While Mireshghallah et al. 2020 does add noise, it does not offer any differential privacy guarantee.
- Dropout Defense for preventing reconstruction of visual data has been explored in the context of convolution channel obfuscation [1].
- L249-250 assert that existing techniques do not leverage feature level data. I think this may not be true w.r.t the attacks used in [1].
- This paper does not compare with other classes of defense techinques such as [2,3,4,5,6].


1. Singh, A., Chopra, A., Garza, E., Zhang, E., Vepakomma, P., Sharma, V. and Raskar, R., 2021. DISCO: Dynamic and Invariant Sensitive Channel Obfuscation for deep neural networks. In Proceedings of the IEEE/CVF Conference on Computer Vision and Pattern Recognition (pp. 12125-12135).
2. Rezaeifar, S., Voloshynovskiy, S., Asgari Jirhandeh, M. and Kinakh, V., 2022. Privacy-Preserving Image Template Sharing Using Contrastive Learning. Entropy, 24(5), p.643.
3. Osia, S.A., Shamsabadi, A.S., Sajadmanesh, S., Taheri, A., Katevas, K., Rabiee, H.R., Lane, N.D. and Haddadi, H., 2020. A hybrid deep learning architecture for privacy-preserving mobile analytics. IEEE Internet of Things Journal, 7(5), pp.4505-4518.
4. Vepakomma, P., Singh, A., Gupta, O. and Raskar, R., 2020, November. NoPeek: Information leakage reduction to share activations in distributed deep learning. In 2020 International Conference on Data Mining Workshops (ICDMW) (pp. 933-942). IEEE.
5. Li, A., Guo, J., Yang, H., Salim, F.D. and Chen, Y., 2021, May. DeepObfuscator: Obfuscating intermediate representations with privacy-preserving adversarial learning on smartphones. In Proceedings of the International Conference on Internet-of-Things Design and Implementation (pp. 28-39).
6. Liu, S., Du, J., Shrivastava, A. and Zhong, L., 2019. Privacy adversarial network: representation learning for mobile data privacy. Proceedings of the ACM on Interactive, Mobile, Wearable and Ubiquitous Technologies, 3(4), pp.1-18.

---

> ### Author Response · Authors · 2022-08-02
> **Response to Reviewer d4JB**
>
> *References*:
>
> [1] Singh, A., Chopra, A., Garza, E., Zhang, E., Vepakomma, P., Sharma, V. and Raskar, R., 2021. DISCO: Dynamic and Invariant Sensitive Channel Obfuscation for deep neural networks. In Proceedings of the IEEE/CVF Conference on Computer Vision and Pattern Recognition (pp. 12125-12135).
>
> [2] Rezaeifar, S., Voloshynovskiy, S., Asgari Jirhandeh, M. and Kinakh, V., 2022. Privacy-Preserving Image Template Sharing Using Contrastive Learning. Entropy, 24(5), p.643.
>
> [3] Osia, S.A., Shamsabadi, A.S., Sajadmanesh, S., Taheri, A., Katevas, K., Rabiee, H.R., Lane, N.D. and Haddadi, H., 2020. A hybrid deep learning architecture for privacy-preserving mobile analytics. IEEE Internet of Things Journal, 7(5), pp.4505-4518.
>
> [4] Vepakomma, P., Singh, A., Gupta, O. and Raskar, R., 2020, November. NoPeek: Information leakage reduction to share activations in distributed deep learning. In 2020 International Conference on Data Mining Workshops (ICDMW) (pp. 933-942). IEEE.
>
> [5] Li, A., Guo, J., Yang, H., Salim, F.D. and Chen, Y., 2021, May. DeepObfuscator: Obfuscating intermediate representations with privacy-preserving adversarial learning on smartphones. In Proceedings of the International Conference on Internet-of-Things Design and Implementation (pp. 28-39).
>
> [6] Liu, S., Du, J., Shrivastava, A. and Zhong, L., 2019. Privacy adversarial network: representation learning for mobile data privacy. Proceedings of the ACM on Interactive, Mobile, Wearable and Ubiquitous Technologies, 3(4), pp.1-18.
>
> [7] He, Z., Zhang, T., & Lee, R. B. (2019, December). Model inversion attacks against collaborative inference. In Proceedings of the 35th Annual Computer Security Applications Conference (pp. 148-162).
>
> [8] Xiao, T., Tsai, Y. H., Sohn, K., Chandraker, M., & Yang, M. H. (2020, April). Adversarial learning of privacy-preserving and task-oriented representations. In *Proceedings of the AAAI Conference on Artificial Intelligence* (Vol. 34, No. 07, pp. 12434-12441).

---

> ### Author Response · Authors · 2022-08-02
> **Response to Reviewer d4JB**
>
> We greatly appreciate the invaluable and constructive comments from the reviewer. We address each of the concerns below.
>
> **Question:** *“How is your contribution 1 (L53-55) different from contribution 3 (L59-63)? I believe both refer to the experimental evaluation only.”*
>
> **A:** We agree that our description of contribution 1 is not very precise. In fact, our contribution 1 is to measure the effectiveness of existing defence mechanisms against reconstruction attacks to demonstrate the limitations of existing defence mechanisms. But, in contrast, our contribution 3 is to evaluate the effectiveness of our proposed anti-defence method comparing with the baseline. We have revised the description of our contribution 1 as follows: "We systematically measure the effectiveness of three state-of-the-art defence mechanisms against reconstruction attacks in edge-cloud collaborative inference systems. We show that the obfuscated representations still retain the latent privacy features that can be extracted to further optimize attacks.".
>
> **Weakness1:** *“The paper shouldn't call the attack techniques as model inversion since the goal is to infer sensitive information from the data used for prediction query and not the training data used. ”*
>
> **A:** We thank the reviewer for this comment. We follow the concept, model inversion, used by [7] to make a comparison with [7]. We agree that it is more appropriate to call the attack techniques as reconstruction attacks. We will revise it in our manuscript accordingly.
>
> **Weakness2:** *“Wrong characterization of early MI attacks by Fredrikson et al. 2014 and Fredrikson et al. 2015 that extract sensitive attributes (Fredrikson et al. 2014) and reconstruction of original data (Fredrikson et al. 2015).”*
>
> **A:** We agree that our descriptions of early MI attacks by Fredrikson et al. 2014 and Fredrikson et al. 2015 are inaccurate. We will clarify them in our manuscript.
>
> **Weakness3:** *“While Mireshghallah et al. 2020 does add noise, it does not offer any differential privacy guarantee.”*
>
> **A:** Thanks for the comment. We will rephrase this sentence to: "Mireshghallah et al. 2020 adopts a scheme that adds Laplacian noise to protect the privacy of inference data".
>
> **Weakness4:** *“Dropout Defense for preventing reconstruction of visual data has been explored in the context of convolution channel obfuscation [1].”*
>
> **A:** Reference [1] is inspiring work and we’re sorry for missing discussing it in our paper. However, there are several differences between [1] and our work. First, the Random Channel Pruning defense evaluated in [1] is close to the dropout defense, but [1] uses the likelihood maximization attack scheme to test the effectiveness of Random Channel Pruning defense. Since the likelihood maximization attack scheme requires performing backpropagation using the weights of the client network, it does not appropriate for the black-box attack setting in our paper. Second, [1] proposes attack and defense schemes for sensitive inputs and attributes, whereas our work focuses on using the supervised decoder to systematically measure and analyze the effectiveness of different defenses. We will cite [1] and explain the above differences clearly in our manuscript.
>
> **Weakness5:** *“L249-250 assert that existing techniques do not leverage feature-level data. I think this may not be true w.r.t the attacks used in [1].”*
>
> **A:** We agree that our assertion in L249-250 may not be true since one of our baselines [7] has already leveraged feature-level data. We will modify the description in **Compared Baseline** and further elaborate on both [7] and [1] in our manuscript.
>
> **Weakness6:** *“This paper does not compare with other classes of defense techniques such as [2,3,4,5,6].”*
>
> **A:** We thank the reviewer for this insightful suggestion and agree that a more comprehensive comparison will improve the quality of our work. We will provide additional comparison experiments or include discussions with respect to [2,3,4,5,6] in our manuscript.
>
> The following are our thoughts regarding those defense techniques:
>
> [2] uses contrastive learning to defend against reconstruction attacks and attribute inference attacks. As this work was published in the same month when our paper was submitted, we unfortunately missed discussing it.
>
> [3] combines dimensionality reduction, noise injection, and Siamese fine-tuning to protect sensitive information from features, but the main goal of [3] is to defend against private attribute leakage, not reconstruction attacks.
>
> [4] focuses on training data privacy. However, our work focuses on inference data privacy.
>
> Both [5] and [6] are adversarial training frameworks that can simultaneously defend against both reconstruction attacks and private attribute leakage. Their min-max optimization functions against reconstruction attacks are very close to the Adversarial Learning defense [8] evaluated in our paper.

---

> > ### Comment · Reviewer_d4JB · 2022-08-08
> > **Response to the Authors**
> >
> > I thank the authors for clarifying my concerns.
> >
> > Based on the author's response to my "question" it appears to me that both contributions are essentially the same. The authors introduce a new attack technique SFD and they perform an experimental evaluation of their attack. They compare SFD with an existing attack technique and evaluate the two against existing defense techniques.
> >
> > My overall take on this work is that the contribution is limited to a new modification to the existing feature inversion attack (SFD). While the authors have done a good job in empirical evaluation, I believe this work perhaps might be more suitable for a benchmarking effort. In the light of the author's response, I have improved the score but I still think this work has not crossed the acceptance threshold in my opinion.
> >
> > Minor feedback -
> >
> > **Weakness 4** : As the authors pointed out themselves, [1] does have a supervised decoder attack as one of the evaluations.
> >
> > **Weakness 6**: [3] does have a reconstruction attack evaluated in Fig. 11. For [2], my apologies, I didn't realize the timing. For [4], I believe the work is proposed for both training and inference. There is some citation mismatch that I was able to find here [8]. For [5] and [6], I agree. The only key difference between the two is in the choice of reconstruction loss metric. Since DeepObfuscator uses SSIM loss.
> >
> >
> > 9. Vepakomma, P., Singh, A., Zhang, E., Gupta, O. and Raskar, R., 2021, December. NoPeek-Infer: Preventing face reconstruction attacks in distributed inference after on-premise training. In 2021 16th IEEE International Conference on Automatic Face and Gesture Recognition (FG 2021) (pp. 1-8). IEEE.

---

> > > ### Author Response · Authors · 2022-08-09
> > > **Further clarification on our major contributions**
> > >
> > > Thank you very much for your follow-up comments. We would like to further clarify our major contributions.
> > >
> > > As indicated by our title, "Measuring Model Inversion Defences in Edge–Cloud Collaborative Inference Systems", the major goal of this work is to take the first step toward **measuring** the robustness of the state-of-the-art defense countermeasures with respect to MI attacks for edge-cloud collaborative inference systems. Our measurement results demonstrate that those existing defenses are not robust enough, and we hope our findings can encourage researchers to pursue more robust defence mechanisms against MI attacks.
> > >
> > > To make our contributions much clearer, we would like to rewrite our *contribution 1* and *contribution 3*, respectively, as follows:
> > >
> > > * We present the privacy and utility trade-offs of the state-of-the-art defence mechanisms against the Existing Model Inversion (EMI) attack. We show that the obfuscated representations still retain the latent privacy features that can be extracted to further optimize attacks.
> > >
> > > * We conduct a comprehensive measurement of the robustness of three representative image classification tasks leveraging our proposed anti-defence technique. Our experiments show that, in the worst case, our technique can achieve 67.03% improvement in the attack performance compared with the EMI attack, making the existing defenses significantly less effective. We hope our findings inspire further work in increasing the robustness of defence mechanisms against MI attacks for edge-cloud collaborative inference systems.

---

### Official Review · Reviewer_HyYW · 2022-08-04

**Rating:** 5
**Confidence:** 3
**Soundness:** 2 fair
**Presentation:** 4 excellent
**Contribution:** 3 good

**Summary:**

This paper looks into the three existing privacy protection mechanisms for edge-cloud collaborative inference systems and proposes an anti-defense method, dubbed Sensitive Feature Distillation (SFD), which simultaneously uses two existing techniques, shadow model and feature-based knowledge distillation, for attacking the privacy protection mechanisms. The core idea is to develop a shadow model for the attack-targeting model, which can produce the non-obfusticated and labeled training data for the privacy-protecting model, and then feeds this model to the knowledge distillation process to imitate or learn the behavior of the target model, so that the resulting model can produce the non-obfusticated (or purified) feature map, which can lead to the privacy leak. The authors claim that the proposed attack method can effectively extract the purified feature map from the intentionally obfusticated one and recover the private image for popular image datasets such as MNIST, CIFAR10, and CelebA.

**Questions:**

* Currently, the manuscript is written in such a way that there are various assumptions that seem to make the proposed technique work. Moreover, the experiments are performed in quite specific setups, which makes me question about the generality and practicality of proposed solution. Could you please elaborate on this?
* Especially, the model split point should play a significant role to evaluate the effectiveness of proposed solutions and make comparisons with the existing techniques. I believe, to be fair, the split points should be similar to what prior works used. Could you please comment on this?
* If I understood correctly, the attackers not only have model's behavior (functionality), model architecture, and hyperparameters a priori, but also have the labeled training data for ground truth that includes possible image inputs and corresponding intermediate feature map results. Should the attackers have all these conditions satisfied to try an attack, or are some of these conditions optional?

**Limitations:**

Elaborated above in the weaknesses part.

**Strengths And Weaknesses:**

Strengths
* This paper tackles an interesting problem, the privacy protection in edge-cloud collaborative systems.
* This paper identifies limitations of the existing solutions, which seem valid issues.
* Writing quality is sufficient for nonknowledgable readers to follow and comprehend the idea.

Weaknesses
* The proposed technique is based on narrow assumptions that much information is already available to the attackers and given. The examples are the model architecture, hyperparameters, and training data for shadow model (Ds). Moreover, the proposed technique narrows the target data type to be images.
* The experiments focus on very early split points such as 1st or 2nd conv layer or 1st fc layer. This settings are favorable to the proposed technique since the existing techniques would be weaker to the model inverse attacks. On the other hand, prior works do not seem to split the model in such an early point in the model architecture. For instance, Table 2 in [Mireshghallah et al., 2020] reports that they run 4~11 layers in the edge while offloading the rest to the cloud. For fair comparison, I believe the methodology should be consistent among the baselines and proposed system.
* While Section 5 offers the analysis for the implication of shadow model, the experiments still use quite small, simple, and outdated models such as VGG-13, VGG-16, and ResNet-18, which do not seem adequate for demonstrating the robustness on the different model architectures.

---

> ### Author Response · Authors · 2022-08-06
> **Response to Reviewer HyYW**
>
> *References*:
>
> [1] Pal, S., Gupta, Y., Shukla, A., Kanade, A., Shevade, S., & Ganapathy, V. (2020, April). Activethief: Model extraction using active learning and unannotated public data. In *Proceedings of the AAAI Conference on Artificial Intelligence* (Vol. 34, No. 01, pp. 865-872).
>
> [2] Oh, S. J., Schiele, B., & Fritz, M. (2019). Towards reverse-engineering black-box neural networks. In *Explainable AI: Interpreting, Explaining and Visualizing Deep Learning* (pp. 121-144). Springer, Cham.
>
> [3] Xiao, T., Tsai, Y. H., Sohn, K., Chandraker, M., & Yang, M. H. (2020, April). Adversarial learning of privacy-preserving and task-oriented representations. In *Proceedings of the AAAI Conference on Artificial Intelligence* (Vol. 34, No. 07, pp. 12434-12441).
>
> [4] Titcombe, T., Hall, A. J., Papadopoulos, P., & Romanini, D. (2021). Practical defences against model inversion attacks for split neural networks. *arXiv preprint arXiv:2104.05743*.
>
> [5] He, Z., Zhang, T., & Lee, R. B. (2020). Attacking and protecting data privacy in edge–cloud collaborative inference systems. *IEEE Internet of Things Journal*, *8*(12), 9706-9716.
>
> [6] Liu, Z., Mao, H., Wu, C. Y., Feichtenhofer, C., Darrell, T., & Xie, S. (2022). A convnet for the 2020s. In *Proceedings of the IEEE/CVF Conference on Computer Vision and Pattern Recognition* (pp. 11976-11986).

---

> ### Author Response · Authors · 2022-08-06
> **Response to Reviewer HyYW**
>
> **Weakness1:** *“The proposed technique is based on narrow assumptions that much information is already available to the attackers and given. The examples are the model architecture, hyperparameters, and training data for shadow model (Ds). Moreover, the proposed technique narrows the target data type to be images.”*
>
> **A:** Please refer to our anwser for **Question3**. We thank the reviewer for the suggestion on the target data type, we will further discuss the feasibility of extending the proposed technique to other target data types in our revised manuscript.
>
> **Weakness2:** *“The experiments focus on very early split points such as 1st or 2nd conv layer or 1st fc layer. This settings are favorable to the proposed technique since the existing techniques would be weaker to the model inverse attacks. On the other hand, prior works do not seem to split the model in such an early point in the model architecture. For instance, Table 2 in [Mireshghallah et al., 2020] reports that they run 4~11 layers in the edge while offloading the rest to the cloud. For fair comparison, I believe the methodology should be consistent among the baselines and proposed system.”*
>
> **A:** Our experiments did not focus on very early split points. **Table 1** in our paper reports the split point settings of different target models. Taking VGG-16 as an example, 3rd conv block means the 7th conv layer of the model (13 conv + 1 fc). Similarly, 4th conv block means the 49th conv layer for ResNet-50 model (49 conv + 2 fc). Therefore, our experiments cover the setting of split points at different depths of the model. We didn't show the experimental results of deeper split points (such as the 4th conv block) in VGG-16 because the attack results of EMI are already very poor (Even without any defense, the attacker can't reconstruct recognizable input data. Under such a split point setting, it is unnecessary to deploy defense in the system).
>
> The settings of our experiments are consistent with the baselines. The baselines we compared are [3,4,5]. On the MNIST dataset, the split points set by the baseline are the 1st and 2dn conv layer. On the CIFAR10 dataset, the split points set by the baseline are the 1st and 2nd conv block. On the CelebA dataset, the split points set by the baseline are the 2nd, 3rd, 4th conv block, avgpool and 1st fc layer. Our experimental setups therefore cover the baseline setups.
>
> **Weakness3:** *“While Section 5 offers the analysis for the implication of shadow model, the experiments still use quite small, simple, and outdated models such as VGG-13, VGG-16, and ResNet-18, which do not seem adequate for demonstrating the robustness on the different model architectures.”*
>
> **A:** Thanks for the comment. We agree that using newer and more complex models (e.g. ConvNext [6]) for such an experiment would be more helpful to demonstrate its robustness. We will provide additional comparison experiments in revised version of our manuscript.

---

> ### Author Response · Authors · 2022-08-06
> **Response to Reviewer HyYW**
>
> We greatly appreciate the invaluable and constructive comments from the reviewer. We address each of the concerns below.
>
> **Question1:** *“Currently, the manuscript is written in such a way that there are various assumptions that seem to make the proposed technique work. Moreover, the experiments are performed in quite specific setups, which makes me question about the generality and practicality of proposed solution. Could you please elaborate on this?”*
>
> **A:** We thank the reviewer for the comment. We think the generality and practicality of our proposed solution are reasonable. The main goal of our paper is to study the potential privacy leakage points of existing defense solutions in edge-cloud collaborative inference systems. In such a system, appropriate split points are set at different depths of the model in order to maintain the trade-off between privacy and computational efficiency. Therefore, in our experimental setup, different architectures of the target model are used for three standard benchmark datasets, as well as the split points at different depths of the target model for shallow, medium and deep layers respectively. Our experimental setup is also a fairly general one. For the attacker’s capability, we assume that the attacker has the same distribution dataset as the training data of the target model and the knowledge of the target model architecture. As for the attacker's dataset, this is the standard setting for performing MI attacks. As for the architecture knowledge, in order to achieve this in practice, the adversary can perform model extraction attacks and model hyper-parameter stealing attacks [1,2].
>
> **Question2:** *“Especially, the model split point should play a significant role to evaluate the effectiveness of proposed solutions and make comparisons with the existing techniques. I believe, to be fair, the split points should be similar to what prior works used. Could you please comment on this?”*
>
> **A:** We thank the reviewer for the comment. In order to make a comprehensive and fair comparison, our setting of the split points is similar to the prior works. More details please refer to our answer for **Weakness2**.
>
> **Question3:** *“If I understood correctly, the attackers not only have model's behavior (functionality), model architecture, and hyperparameters a priori, but also have the labeled training data for ground truth that includes possible image inputs and corresponding intermediate feature map results. Should the attackers have all these conditions satisfied to try an attack, or are some of these conditions optional?”*
>
> **A:** The attackers only need to satisfy a part of these conditions to try the attack. We assume that the attacker has a local dataset comes from the same distribution as the target model’s training set, and also assume that the attacker has the knowledge of the target model architecture and hyperparameters (see L151-167 of our paper). The attacker does not need to know the behavior of the target model, and can train the shadow model for different tasks. At the same time, the assumption that the attacker has knowledge of the target model architecture can be further relaxed (See L320-329 of our paper).

---

### Meta-Review · Area_Chair_V4xm · 2022-08-26

**Recommendation:** Accept
**Confidence:** Less certain

**Metareview:**

The paper presents a new model inversion attack for edge-cloud learning which is shown to overcome existing defenses against model inversion in such scenario. The attack is based on Sensitive Feature Distillation (SFD) which simultaneously uses two existing techniques, shadow model and feature-based knowledge distillation. The authors claim that the proposed attack method can effectively extract the purified feature map from the intentionally obfuscated one and recover the private image for popular image datasets such as MNIST, CIFAR10, and CelebA.
The proposed method is quite novel and its evaluation is solid. The method is based on certain assumptions and utilizes several techniques from related work in order to meet these assumptions, which makes the contribution rather incremental. The paper is well-written and the relationship to existing work is clearly presented and discussed.

**Award:**

No

---

### Decision · Program_Chairs · 2022-09-14

Accept